# Mechanically Robust Gastroretentive Drug-Delivery Systems Capable of Controlling Dissolution Behaviors of Coground β-Lapachone

**DOI:** 10.3390/pharmaceutics11060271

**Published:** 2019-06-10

**Authors:** Hyeongmin Kim, Chung-Lyol Lee, Seohyun Lee, Tae Jin Lee, Iqra Haleem, Younghong Lee, Na Jung Hwang, Kyusun Shim, Dohyun Kim, Jaehwi Lee

**Affiliations:** College of Pharmacy, Chung-Ang University, Seoul 06974, Korea; hm.kim8905@gmail.com (H.K.); leechung22@naver.com (C.-L.L.); krxid@naver.com (S.L.); pelotas@hanmail.net (T.J.L.); iqra.haleem24@gmail.com (I.H.); iuyhyj@naver.com (Y.L.); zoeyoj@naver.com (N.J.H.); kssim95@naver.com (K.S.); dylan13@naver.com (D.K.)

**Keywords:** swelling gastroretentive system, cogrinding technique, freeze–thawing method, β-lapachone, dissolution behavior, mechanical strength

## Abstract

In this study, we aimed to design a highly swellable and mechanically robust matrix tablet (SMT) as a gastroretentive drug-delivery system (GRDDS) capable of improving the dissolution behavior of β-lapachone with low aqueous solubility. For the preparation of SMTs, the cogrinding technique and freeze–thaw method were used to disperse β-lapachone in SMTs in an amorphous state and to enhance the swelling and mechanical properties of SMTs, respectively. As a result, the crystallinity of coground β-lapachone incorporated in the SMTs was found to be considerably decreased; thereby, the dissolution rates of the drug in a simulated gastric fluid could be substantially increased. The SMTs of β-lapachone also demonstrated significantly enhanced swelling and mechanical properties compared to those of a marketed product. The reason for this might be because the physically crosslinked polymeric networks with a porous structure that were formed in SMTs through the freeze–thaw method. In addition, β-lapachone was gradually released from the SMTs in 6 h. Therefore, SMTs of β-lapachone developed in this study could be used as GRDDS with appropriate swelling and mechanical properties for improving the dissolution behavior of hydrophobic drugs such as β-lapachone.

## 1. Introduction

The oral route has been the most preferred drug-administration route owing to its ease of administration, good patient compliance, and cost effectiveness. To maximize the advantages of oral drug administration, various micro-/nanoparticulate systems for controlled drug release have been extensively explored [1]. However, such micro-/nanoparticulate systems may not be effective in obtaining the high oral bioavailability of drugs with narrow absorption sites, or unstable or poorly soluble natures in alkaline pH [2]. To increase the oral bioavailability of such drugs, gastroretentive drug-delivery systems (GRDDS) can be useful. GRDDS can stay in the stomach and release drugs for a prolonged period, thereby enhancing the oral bioavailability and therapeutic efficacy of the drugs [3]. 

Various types of GRDDS with different gastroretentive mechanisms have been investigated, such as floating [4], high-density [5], muco-adhesive [6], raft-forming [7], magnetic [8], and expandable systems [9]. The most extensively explored types of GRDDS are floating or expandable systems. They can float in the gastric fluid or increase their volume through absorbing the surrounding fluid, thereby residing in the stomach for a sufficient period of time. Owing to their robust gastroretentive performance and simple manufacturing procedure compared to other types of GRDDS, floating and expandable GRDDS have been globally commercialized by many pharmaceutical companies [10]. 

However, as with other types of GRDDS, floating and expandable GRDDS have shown considerable limitations. For instance, floating GRDDS indispensably need high gastric-fluid levels to appropriately float and work [10]. In the case of expandable GRDDS, it is difficult for them to maintain an expanded state without losing their structural integrity [11]. In addition, most existing GRDDS have not been specially formulated to solubilize hydrophobic drugs and improve the dissolution behavior of the drugs; therefore, they have been generally used only for hydrophilic drugs [12,13,14,15]. 

To overcome the limitations of existing GRDDS, in this study we aimed to develop a mechanically robust swellable matrix tablet (SMT) capable of improving the dissolution behavior of hydrophobic drugs. For promoting the dissolution of hydrophobic drugs incorporated in SMT, cogrinding technology was used to prepare SMTs. Cogrinding hydrophobic drugs, hydrophilic polymers, and solid surfactants can reduce the crystallinity of the drugs, thereby increasing the aqueous solubility and dissolution rate of the drugs [16]. For prolonged gastric retention, SMTs also need to exhibit suitable swelling behavior with a robust mechanical property when exposed to the gastric fluid. To this end, the freeze–thaw method was exploited to prepare the SMTs because polymers constituting SMTs can be physically crosslinked by a repetitive freeze–thawing procedure, thereby presenting enhanced mechanical strength without considerable changes in their swelling property [17,18,19]. As a model drug, β-lapachone, a bioactive naphthoquinone, was used because it shows low aqueous solubility (~55 μg/mL) [20] and instability under the basic conditions of the small and large intestine due to its sensitivity to alkaline hydrolysis [21,22]. 

## 2. Materials and Methods 

### 2.1. Materials

β-Lapachone was supplied by the Mazence Pharmaceutical Company (Suwon, Korea). Crospovidone (Kollidon CL), poloxamer 407 (Lutrol F 127), and poloxamer 188 (Lutrol F 68) were obtained from BASF (Ludwigshafen, Germany). Poly(acrylic acid) (Carbopol 940) was procured from Lubrizol Advanced Materials Inc. (Cleveland, OH, USA). Xanthan gum, hydroxypropyl methylcellulose (HPMC), sucrose, and poly(ethylene oxide) (PEO) were purchased from the Sigma-Aldrich Company (St. Louis, MO, USA). Polyethylene glycol (PEG) 4000 was obtained from Daejung Chemicals and Metal Co., Ltd. (Siheung, Korea). Polydextrose was purchased from Santa Cruz Biotechnology Inc. (Dallas, TX, USA). Spray dried lactose was kindly provided by Whanin Pharm Co., Ltd. (Suwon, Korea). All other materials used were of pharmaceutical grade.

### 2.2. Preparation of Coground Mixtures of β-Lapachone, Hydrophilic Materials, and Sodium Lauryl Sulfate

Coground mixtures of β-lapachone, hydrophilic materials (poloxamer 188, poloxamer 407, sucrose, polydextrose, polyethylene glycol (PEG) 4000, or hydroxypropyl methylcellulose (HPMC)) and sodium lauryl sulfate (SLS) were prepared using a laboratory ball mill (UBM-200S, Intec System Co. Ltd., Seongnam, Korea). The compositions of the coground mixtures are presented in Table 1. β-lapachone was first mixed with each hydrophilic material and SLS using a vortex mixer for 5 min. The mixtures were then put into 500 mL ball mill pots containing balls with different diameters of 10, 13, and 20 mm, and coground for 24 h. 

### 2.3. In Vitro Dissolution Study of Coground Mixtures

In vitro dissolution study of the coground mixtures and raw β-lapachone powder was conducted using USP dissolution apparatus II. The raw β-lapachone powder was passed through an 80-mesh sieve before conducting the dissolution study, and used without being tabletized. Samples equivalent to 40 mg of β-lapachone were added to 900 mL simulated gastric fluid (pH 1.2) containing SLS at a concentration of 0.5% (*w*/*v*) and stirred at 50 rpm and 37 °C. Aliquots (1 mL) were withdrawn at predetermined time points (5, 10, 15, 30, 60, 90, and 120 min) and the dissolution vessels were replenished with the equal volume of fresh dissolution medium. The aliquots were filtered through a 0.45 µm polyvinylidene fluoride syringe filter, and the levels of β-lapachone in each sample were analyzed with a high-performance liquid chromatography (HPLC) system (Breeze 2 HPLC system, Waters Corporation, Milford, MA, USA) based on a previously validated analysis method [23]. VARIAN Polaris^TM^ C_18_-A column (150 × 4.6 mm, 5 μm particle size, Agilent Technologies, Inc., Santa Clara, CA, USA) was used, and a mobile phase consisting of 32% distilled water and 68% methanol flowed at a rate of 1 mL/min. The detection wavelength was set at 254 nm.

### 2.4. X-Ray Powder Diffraction (XRD) Analysis 

XRD patterns of the coground mixtures were evaluated using an X-ray powder diffractometer (New D8-Advance, Bruker-AXS, Billerica, MA, USA). Cu–Kα radiation was generated at 30 mA and 40 kV. Scanning speed was 2θ/s, and the data were analyzed by continuous mode in a range of 5°–50° at a step size of 0.02°.

### 2.5. Preparation of Freeze–Thawed Polymer Mixtures (FPM)

Crospovidone, poly(acrylic acid), and xanthan gum were passed through a 40-mesh sieve, respectively, and mixed at a weight ratio of 1:1:1. The 3 different hydrophilic polymers were selected because they were known to exhibit good swellable properties [24,25,26] The 1:1:1 weight ratio of the polymers was selected because in our preliminary experiment it demonstrated the best swelling ability among the FPM composed of the polymers at various weight ratios. The mixtures were then added to distilled water to prepare 5% (*w*/*w*) solution of the mixtures. The samples were thinly spread on plastic weighing dishes and frozen at –20 °C for 3 h. After the freezing process, they were thawed at room temperature (~22 °C) for 3 h. The freeze–thaw cycle was repeated 3 times. The FPM were then dried in an oven at 40 °C. The dried samples were finally ground using a pulverizer.

### 2.6. FPM Scanning Electron Microscopy

The surface morphology of FPM and non-FPM composed of the same polymers was observed using a scanning electron microscope (SEM) (Model S-3400N, Hitachi Ltd., Tokyo, Japan). The samples were mounted on a double-sided carbon tape attached to a metal stub and sputter-coated with platinum. The samples were then observed at an accelerating voltage of 10.0 kV.

### 2.7. Preparation and Water Uptake Study of FPM Matrices

FPM-based matrices were prepared using a hydraulic press equipped with a 13 mm diameter flat-faced tooling and a single-punch tablet machine (Riken Power, Riken Seiki Co., Ltd., Niigata, Japan). The total weight of FPM matrices was fixed at 1000 mg. To evaluate the effect of different weight ratios of FPM in the matrices on their swelling property, FPM matrices were prepared using FPM at different weight ratios of 20–80%. The FPM matrices contained PEO at a weight proportion of 20% to enhance the mechanical strength of the matrices in swollen state. FPM matrices incorporated FPM at weight ratios of 20–60%, and spray-dried lactose was added to them to equalize the weight of the matrices to be 1000 mg. All the mixtures of FPM and other excipients were compressed at 200 kgf/cm^2^ using the hydraulic press to obtain FPM-based matrices.

Weighed FPM matrices (W_0_) were placed in a dissolution tester vessel filled with 900 mL pH 1.2 simulated gastric fluid and agitated at 50 rpm and 37 °C. At predetermined time intervals (5, 10, 15, 30, 60, 90, and 120 min), the FPM matrices were withdrawn from the medium, and water on the surface of the matrices was carefully removed with a paper towel. The weight of the swollen matrices (W_s_) was then measured. The swelling index of each sample was calculated using the following equation [27]:$$\mathrm{Swelling}\;\mathrm{index}\;\left(\%\right)\;=\frac{W_{s}{-W}_{0}}{W_{0}}\times \;100$$

### 2.8. Preparation of SMT of β-Lapachone

SMTs of β-lapachone were prepared as presented in Table 2. All constituents of the SMTs were passed through a 60-mesh sieve and thoroughly mixed in a plastic bag. The mixtures were tabletized using the hydraulic press as described above. As a control group, a marketed gastroretentive swelling tablet (DepoMed Inc., Newark, CA, USA) was used [28].

### 2.9. Water-Uptake Study of SMT of β-Lapachone

The water-uptake study of the SMTs prepared was performed in accordance with the same procedure as described above.

### 2.10. Assessment of SMT Mechanical Strength

The mechanical strength of the SMTs was evaluated through force-displacement measurement using a texture analyzer (TA plus, Lloyd instrument Ltd., Bognor Regis, UK). Prior to measurement, the SMTs were exposed to pH 1.2 simulated gastric fluids under the same condition of the aforementioned drug-dissolution test. At predetermined time intervals (1, 2, 3, 4, 5, 6, 7, and 8 h), the swollen tablets were taken out and gently patted with a tissue paper to remove water droplets. The swollen tablets were then compressed with a flat-tipped stainless steel probe of 10 mm in diameter and 70 mm in height at a displacement speed of 2 mm/min. The force at 3 mm displacement was measured. 

### 2.11. In Vitro Drug0Release Study of β-Lapachone SMTs

An in vitro drug-release study was performed to evaluate the drug-dissolution behavior of the SMTs in accordance with the same procedure of the drug-dissolution test as described above, except aliquots of 1 mL were withdrawn from the dissolution media at predetermined time intervals of 0.5, 1, 1.5, 2, 3, 4, 5, 6, 7 and 8 h.

To investigate the effects of SMT composition on drug-release kinetics, drug-dissolution data were fitted to various mathematical equations of drug release: zero-order model (Equation (1)), first-order model (Equation (2)), Higuchi model (Equation (3)), and Korsmeyer–Peppas model (Equation (4)).
(1)$$M_{t}=M_{0}+k_{0}t$$
(2)$${\mathrm{logC}}_{t}={\mathrm{logC}}_{0}-\frac{k_{1}t}{2.303}$$
(3)$$M_{t}={\;k}_{2}\sqrt{t}$$
(4)$$\log\left(\frac{M_{t}}{M_{\infty }}\right)=\log k_{3}+\mathrm{nlogt}$$
where M_t_ is the amount of the drug at time t; M_0_ is the initial amount of the drug incorporated in the SMTs; C_t_ is the concentration of the drug in dissolution media at time t; C_0_ is the initial concentration of the drug in dissolution media at time; M_t_/M_∞_ is the fraction of the drug released at time t; *n* is the release exponent of Korsmeyer–Peppas model; and k_0_, k_1_, k_2_, and k_3_ are the zero-order, first-order, Higuchi, and Korsmeyer-Peppas rate constants, respectively [29].

### 2.12. Statistical Analysis

All experiments were conducted in triplicate. Means were compared by one-way analysis of variance and Student’s t-test. *p* < 0.05 was considered significant.

## 3. Results and Discussion

### 3.1. In Vitro Dissolution Behavior and XRD Spectra of β-Lapachone Incorporated in Coground Mixtures

Figure 1 shows in vitro dissolution profiles of β-lapachone contained in coground mixtures and raw β-lapachone powder. For the whole tested period (2 h), raw β-lapachone powder exhibited only ~5% of the dissolved amount of the drug at the end of the experiment, which might be largely attributed to the low aqueous solubility of the drug (~55 μg/mL) [20]. In contrast, allcoground mixtures exhibited considerably increased drug-dissolution rates, showing dissolved amounts of β-lapachone, ranging from 58% to 84% at 2 h. From the coground mixtures, GM 1, which contained poloxamer 407 as a hydrophilic polymer, presented the most greatly enhanced drug-dissolution behavior (approximately 16-fold increment in the dissolved amount of the drug at 2 h). The reason for the increased drug-dissolution rates demonstrated by the coground mixtures might be because the molecular interactions between the hydrophilic materials, SLS, and β-lapachone that occurred during the cogrinding procedure broke the crystalline drug particles apart, and thereby reduced the crystallinity of β-lapachone [30].

To prove the decreased crystallinity of β-lapachone contained in the coground mixture (GM1), XRD analysis was performed. As presented in Figure 2, raw β-lapachone powder exhibited characteristic peaks, indicating the crystallinity of the drug. The physical mixture of the same components to GM 1 also showed peaks similar to those of raw β-lapachone powder and, therefore, the drug was supposed to be contained in the physical mixture in crystalline state. In contrast, GM1 presented considerably decreased intensities of the characteristic peaks shown by raw β-lapachone powder, implying that the crystallinity of the drug was successfully reduced by the cogrinding technique. However, an amorphous halo was not clearly observed in the XRD pattern of GM1. This result might be because our cogrinding technique did not fully cause perfect amorphization of the drug. The cogrinding technique might not be as efficient as the solid dispersion techniques generally employed to obtain tge amorphous form of a drug within polymeric carriers. Nevertheless, the reason for using this cogrinding technique in this study was because the technique can considerably decrease the crystallinity of β-lapachone through only physical milling, and the appropriate selection of excipients without using any organic solvents is frequently used for solid dispersion. In addition, the cogrinding technique only needs comparatively simple, low-cost equipment and excipients such as a ball mill, surfactants, and polymers, whereas solid dispersion systems require specialized equipment, such as hot-melt extruders and spray dryers. Thus, the cogrinding technique could be used as an alternative to other pharmaceutical technologies to produce solid dispersion systems for improving the dissolution of hydrophobic drugs.

Differential scanning calorimetry (DSC) analysis might be necessary to support the result of the XRD analysis and to understand the effect of temperature on the crystallinity of β-lapachone incorporated in GM1. The reason why DSC analysis was not performed in this study was because previously published studies demonstrated that pure β-lapachone powder in crystalline state did not exhibit any endothermic or exothermic peaks other than its characteristic endothermic peak at 157 °C caused by the drug melting [31,32]. In addition, β-lapachone molecularly dispersed in hydrophilic polymer-based matrices also showed no endothermic or exothermic peaks over temperature ranges generally used for DSC analysis. Based on this, in this study, only XRD analysis was conducted to evaluate the crystallinity of β-lapachone incorporated in the coground mixture.

### 3.2. Water-Uptake Study of FPM Matrices

Figure 3 displays the swelling indices of the FPM matrix and non-FPM matrix. The FPM matrix exhibited considerably greater swelling indices for the tested period than the non-FPM matrix. At the end of the experiment (2 h), the swelling indices of the FPM and non-FPM matrices were evaluated to be approximately 340% and 200%, respectively. The enhanced swelling property of the FPM matrix might be ascribed to the microporous structure of the FPM, which might have been formed during the freeze–thawing process. When the FPM and non-FPM matrices were observed using SEM, the FPM matrix showed a porous and reticular structure, whereas the non-FPM matrix revealed an appearance of a dense matrix without pores (Figure 4). The porous structure of the FPM matrix was considered to be advantageous to promote the infiltration of water molecules into the polymeric matrices. The porous structure of the FPM matrix might enable the FPM matrix to swell at a faster rate and a greater degree than the non-FPM matrix. 

However, the initial burst swelling of the FPM matrix was considered to cause significant decrease in the mechanical strength of the matrix, which would be undesirable for SMTs to be retained in the stomach for a sufficient time. Indeed, the FPM matrix exhibited very weak mechanical strength after being swollen in the simulated gastric fluid, and it was eroded at 15 min after the water uptake study started (data not shown). As mentioned above, the balance between the swelling property and mechanical strength of SMTs is crucial for the performance of SMTs as a swelling GRDDS. Therefore, a mechanical-strength-enhancing agent was needed to enhance the mechanical strength of the FPM matrix in swollen state. PEO was selected as a mechanical-strength-enhancing agent because it has been known to efficiently reinforce mechanical properties of gel-type formulations [33].

### 3.3. Swelling Property of Matrices Incorporating FPM at Different Weight Fractions

To optimize the amount of FPM in SMTs, FPM-based matrices containing FPM at different weight fractions of 20–80% were prepared as tablet form, and their swelling ability was evaluated. As can be seen in Figure 5, in general, the FPM matrices prepared using FPM at higher weight fractions presented greater swelling indices. In particular, the matrices incorporating FPM at weight fractions of 40–80% showed significantly greater swelling indices than that with 20% FPM. However, between the FPM matrices containing FPM at weight fractions of 40%, 60%, and 80%, their swelling behaviors were found to not be considerably different. Therefore, the weight fraction of FPM in SMTs was determined to be 40% because a higher weight ratio of FPM in SMTs might considerably weaken the mechanical strength of the tablets in swollen state.

### 3.4. SMT Swelling Behavior in Simulated Gastric Fluid

Swelling behaviors of SMTs prepared with different compositions (Table 2) were evaluated in pH 1.2 simulated gastric fluid. Figure 6A presents the swelling indices of SMTs 3 and 5, which were prepared using FPM and non-FPM, respectively. For the whole tested period (8 h), SMT 3 exhibited substantially greater swelling indices than SMT 5, implying that the FPM successfully enhanced the swelling ability of SMT. This might be because the porous structure of the FPM could efficiently absorb water compared to non-FPM. In addition, the SMTs containing the FPM were found to quickly expand in 10 min after being exposed to the simulated gastric fluid, showing diameters greater than 3 cm, which is required for swelling GRDDS to retain in the stomach [34]. Thus, FPM was selected to be added to SMTs for enhancing the swelling ability of SMTs.

Thereafter, the swelling behaviors of SMTs containing PEO of different molecular weights were assessed. PEO was used for the preparation of SMTs because it can form dense polymeric networks in aqueous environments and thereby reinforce the mechanical strength of SMTs. Figure 6B shows the swelling indices of SMTs 3, 6, and 7, which were prepared with PEO of 1000, 4000, and 8000 kDa, respectively, and evaluated in comparison with the marketed swelling gastroretentive tablet. In general, the SMTs exhibited greater swelling indices than the marketed product for the tested period. This might be attributed to the presence of FPM contained in the SMTs. Among the SMTs incorporating PEO of different molecular weights, SMTs with PEO of a smaller molecular weight presented higher swelling indices. The swelling indices of the SMTs containing PEO of 1000 (SMT 3), 4000 (SMT 6), and 8000 kDa (SMT 7), evaluated at 6 h, were calculated to be 275.19 ± 19%, 260.15 ± 4.9%, and 250.52 ± 4%, respectively. The reason for this was supposed to be because PEO of a higher molecular weight was more hydrophobic and largely entangled, thereby delaying the diffusion of water molecules into the SMTs and the swelling rate of the tablets [35]. Thus, 1000 kDa PEO was chosen to prepare SMTs for further experiments because it was considered able to enhance the mechanical property of SMTs without causing considerable decrease in the swelling ability of SMTs.

The effect of the weight ratio of 1000 kDa PEO in SMTs on the swelling behavior of SMTs was then examined. Figure 6C illustrates the swelling behavior of SMTs 1–4 incorporating 1000 kDa PEO at different weight fractions of 10%, 15%, 20%, and 30%, respectively, in comparison with the marketed product. SMTs with a lower weight fraction of PEO generally showed a higher swelling ability, and the swelling indices of SMTs 2–4, measured at 8 h, were demonstrated to be higher than that of the marketed tablet (*p* < 0.05). SMT 1 with 10% PEO was observed to have been collapsed at 3 h after starting the experiment. This might be because the amount of PEO contained in SMT 1 was not sufficient to confer a sufficient mechanical strength on SMT 1 required for the tablet to maintain the swollen state under the tested experimental condition. Thus, between the SMTs containing 1000 kDa PEO at different weight ratios, SMT 2 was considered to be the best formulation because it showed suitable swelling and mechanical properties.

### 3.5. Mechanical Strength of SMTs Assessed in Simulated Gastric Fluid

Swelling GRDDS should reside in the stomach for a sufficient period of time, generally longer than 4 h [36,37]. For prolonged gastric retention, swelling GRDDS should endure mechanical stresses generated by gastric peristalsis without being collapsed. To this end, it has been known that they need to exhibit mechanical strength greater than 1.89 N [38]. To confirm that SMTs had appropriate mechanical properties as swelling GRDDS, SMT mechanical strength was evaluated using a texture analyzer compared with the marketed product.

Figure 7 shows the mechanical-strength profiles of the SMTs and the marketed product, evaluated in pH 1.2 simulated gastric fluid for 8 h. The mechanical strength of the SMTs and marketed product decreased as a function of time. This was certainly because the density of the tablets was reduced as tablets were gradually swollen with time. Except for SMT 1, all the tested SMTs maintained mechanical strengths considerably stronger than 1.89 N for more than 6 h without being collapsed, which was calculated based on the measured force values and their diameters in swollen state. Thus, SMTs other than SMT 1 were considered able to withstand the mechanical stresses that occurred by gastric peristalsis and to retain their swollen state. 

Between SMT 3 with FPM and SMT 5 with non-FPM, SMT 3 exhibited considerably weaker mechanical strengths until 4 h than those of SMT 5 (Figure 7A), largely owing to the greater swelling capacity of FPM that that of non-FPM. However, the mechanical strength of SMT 3 measured for the whole tested period was comparable to that of the marketed product. In addition, as aforementioned, the mechanical strength of SMT 3 was also measured to be substantially stronger than 1.89 N. Thus, SMT 3 demonstrated superior swelling property and an appropriate mechanical strength as swelling GRDDS. 

To obtain better SMT mechanical strength, the effect of the molecular weight of PEOs incorporated in SMTs on their mechanical strength was then assessed. As presented in Figure 7B, SMT mechanical strength was found to increase by increasing PEO molecular weight from 1000 (SMT 3) to 8000 kDa (SMT 7). This could be because PEO of a higher molecular weight entangled at a greater degree than that of a lower molecular weight, thereby increasing the density of the SMTs in swollen state [35]. However, the SMTs with PEO of different molecular weights showed similar mechanical strengths 4 h after starting the experiment. The reason for this might be because the SMTs could not maintain the enhanced mechanical strength by the PEO when they were almost fully swollen. Thus, SMT 3, with 1000 kDa PEO, was selected for further experimenting owing to its good swelling property and appropriate mechanical strength. 

The mechanical strength of SMTs prepared using 1000 kDa PEO at different weight ratios (SMTs 1–4) was then evaluated. SMT mechanical strength was found to be enhanced by increasing the PEO weight ratio in the SMTs as shown in Figure 7C. PEO has been known to exhibit a viscoelastic nature in swollen state in an aqueous environment. Thus, SMTs containing a larger amount of PEO were considered to present a stronger mechanical property. SMT 1, which was collapsed in the swelling ability test as described above, exhibited the weakest mechanical strength in all the tested SMTs. Considering that all SMTs except SMT 1 exhibited sufficient mechanical strengths that could endure the mechanical stresses caused by gastric motility, SMT 2 was selected to be the best formulation because it showed the best swelling ability and appropriate mechanical property. 

### 3.6. In Vitro Dissolution Behavior of β-Lapachone Incorporated in SMTs

Although existing swelling GRDDS have demonstrated suitable swelling and mechanical properties, they were generally composed of hydrophilic polymers with a high hygroscopic nature [39]. For this reason, the existing swelling GRDDS are considered to have limitations in solubilizing poorly water-soluble drugs and releasing the drugs at proper rates. To overcome this problem, β-lapachone, a hydrophobic drug with a low aqueous solubility (~55 μg/mL) [20] was coground with poloxamer 407 and SLS, and the coground mixture in which the drug was dispersed in an amorphous state was used for the preparation of SMTs. The weight ratios of SLS and poloxamer 407 used for decreasing the crystallinity of β-lapachone in SMTs were changed from those of GM1 as shown in Table 1 and Table 2. When the same weight ratios of β-lapachone, SLS, and poloxamer 407 to GM1 were used for the preparation of SMTs, the drug dissolution behaviors were not efficiently enhanced (data not shown). The reason for this might be because the weight fraction of SLS in SMTs (2%) was considerably lower than that in GM1 (~11%); thereby, the drug recrystallized while mixing the constituents of SMTs and tableting procedure. Thus, we reoptimized the weight ratios of β-lapachone, SLS, and poloxamer 407 in SMTs to inhibit the recrystallization of β-lapachone during SMT preparation.

In vitro dissolution profiles of β-lapachone contained in SMTs with different compositions are shown in Figure 8. When β-lapachone was added to the SMT as a raw drug powder, only ~10% of the drug was dissolved from the tablet (SMT 8) for the whole tested period (8 h), as presented in Figure 8A. Thus, it was found that β-lapachone could not properly be dissolved from the SMT without an appropriate solubilizing strategy for the drug. In contrast, SMT 3, prepared with GM1 in which β-lapachone was dispersed in amorphous state, exhibited a considerably improved drug dissolution behavior. SMT 3 showed the drug dissolved amount of approximately 60% at 8 h, which was 6 times greater than that assessed with SMT 8. Thus, the coground method was demonstrated to be effective for enhancing the dissolution behavior of β-lapachone from the SMT. The solubility of β-lapachone has been known to not be considerably affected by pH variations [20]. Therefore, the enhanced solubility and dissolution behavior of β-lapachone by the cogrinding technique might be well-maintained in the stomach, and small and large intestine, which is necessary for the efficient absorption of the drug. 

Thereafter, the dissolution behaviors of β-lapachone contained in SMTs with FPM (SMT 3) or non-FPM (SMT 5) were evaluated to examine the effect of the different swelling behaviors of the SMTs on the drug dissolution. As shown in Figure 8B, SMT 3 exhibited a considerably faster drug-dissolution rate than SMT 5. At the end of the experiment, SMT 3 released approximately 60% of β-lapachone, whereas only ~20% of the drug was dissolved in case of SMT 5. In addition, SMT 3 gradually released β-lapachone for the whole tested period, but SMT 5 showed a burst release at 0.5 h, and then the dissolved amount of the drug only slightly increased for the following 7.5 h. In the case of SMT 3, the FPM with a porous structure was supposed to promote the infiltration of water molecules into the tablet, and thereby the dissolution of the amorphous drugs incorporated in SMT 3 could be facilitated. Thus, it was demonstrated that the FPM prepared by the freeze–thaw process could be exploited to enhance the swelling property of SMTs and to improve the dissolution behavior of hydrophobic drugs, such as β-lapachone, from SMTs.

The impact of PEO of different molecular weights on drug-release rates from the SMTs was then examined (SMTs 3, 6, and 7). As presented in Figure 8C, SMT 3 with 1000 kDa PEO exhibited a substantially faster drug-dissolution rate than SMT 6 with 4000 kDa PEO and SMT 7 with 8000 kDa. SMT 3 released ~60% of β-lapachone, whereas SMTs 6 and 7 released less than 40% of the drug for the tested period. This result might also be closely related with the swelling behavior of the tablets. As can be seen in Figure 6B, the tablets prepared with PEO of a lower molecular weight showed greater swelling ability because PEO of a lower molecular weight might be less physically entangled than that of a higher molecular weight, and thereby it was easily swollen. Thus, SMT 3 might have a physically loosened structure compared to SMTs 6 and 7, thereby releasing the incorporated drugs at a faster rate than SMTs 6 and 7. For this reason, 1000 kDa PEO was selected to be used for the preparation of SMTs, because SMTs with 1000 kDa PEO exhibited appropriate mechanical strength, swelling property, and drug-dissolution behavior. 

The effect of the weight ratio of 1000 kDa PEO in SMTs on drug-dissolution behavior was also assessed. As illustrated in Figure 8D, SMTs generally showed faster drug-dissolution rates when the PEO was added to SMTs at a lower weight ratio. In particular, SMT 1 with 10% PEO exhibited a significantly faster drug-release rate (~80% drug release at 8 h) than SMTs 2–4 with 15–30% PEO (approximately 30–70% drug release at 8 h). This might be because PEO contained in SMTs at a lower weight ratio formed a hydrogel with a higher swelling capacity, and thereby released β-lapachone at a faster rate. However, as described above, SMT 1 was collapsed at 3 after being exposed to pH 1.2 simulated gastric fluid because of weak mechanical strength. Therefore, SMT 2 was considered to be the best SMT formulation as it exhibited the most appropriate balance between swelling ability and mechanical strength, along with a desired drug-release profile. 

To examine drug-release kinetics from SMTs, dissolution data were fitted to zero-order, first-order, Higuchi, and Korsmeyer–Peppas models. Correlation coefficients (*R*^2^) for each model evaluated with SMTs of different compositions are presented in Table 3. In general, all SMTs were found to be well-fitted with the zero-order, first-order, Higuchi, and Korsmeyer–Peppas models. However, the Higuchi model can only be applied when the swelling and dissolution of the matrix are negligible [29]. Therefore, we explained the mode of drug release from SMTs by applying zero-order, first-order, and Korsmeyer–Peppas models. The Korsmeyer–Peppas equation is generally used to evaluate drug-release behaviors from polymeric systems when the drug-release mechanism is not well-elucidated, or more than one type of release mechanisms are involved [40]. The release exponent (n) in the Korsmeyer–Peppas equation is used to characterize the drug-release mechanism. If the release exponent is equal to or less than 0.45, drug release is considered to be governed by Fickian diffusion. If the release exponent ranges from 0.45 to 0.89, it indicates anomalous (non-Fickian) transport, which typically involves both diffusion-controlled release and erosion-controlled release. The release exponent of 0.89 denotes Case II (relaxational) transport, and *n* > 0.89 means Super Case II transport [29]. 

SMT 3 with FPM had the best fit with Korsmeyer–Peppas model, whereas SMT 5 with non-FPM exhibited the best fit with the zero-order kinetic. As for SMT 5, its nonporous structure caused by non-FPM could not efficiently absorb water compared to SMT 3 with a porous structure, thereby delaying the dissolution of drug. The *n* value of SMT 3 was determined to be between 0.45 and 0.89 (0.4533), indicating that drug release was governed by both drug diffusion and matrix erosion. The reason for this might be because the presence of FPM in SMT 3 caused a fast swelling rate, thereby promoting the erosion of the polymeric matrix of SMT 3 and the drug release. 

The effect of the molecular weight of PEO on drug-release kinetics from SMTs was then examined. As aforementioned, SMT 3 with 1000 kDa PEO presented a drug-release kinetic that was well-fitted to the Korsmeyer–Peppas model, whereas SMT 6 with 4000 kDa PEO and SMT 7 with 8000 kDa PEO exhibited the best fit with zero-order model. The reason for SMTs 6 and 7 showing zero-order kinetics might be because PEO of a higher molecular weight could be largely entangled and form a physically dense structure, thereby delaying the diffusion of water molecules into the matrices and leading to the slow drug release from SMTs. 

The impact of the weight fraction of PEO in SMTs on drug-release behaviors was then explored. SMTs 1–3 showed the best fit with the Korsmeyer–Peppas model, whereas SMT 4 presented the best fit with the zero-order kinetic. The *n* value determined from SMTs 1–3 ranged from 0.45 to 0.89. The *n* value also decreased by increasing the weight fraction of PEO in SMTs. This might be because the higher weight fraction of PEO might have formed more condensed structures; consequently, drug release was mainly governed by drug diffusion rather than by matrix erosion. In case of SMT 4, the highest fraction of PEO in the tablet was supposed to inhibit the infiltration of water into the polymeric matrix and delay drug release. Thus, SMT 4 showed the zero-order drug-release kinetic.

## 4. Conclusions

In conclusion, hydrophilic polymer-based SMTs of β-lapachone were prepared using the cogrinding and freeze–thaw methods. It was demonstrated that β-lapachone incorporated in the coground mixture exhibited significantly reduced crystallinity; thereby, the dissolution behavior of the drug incorporated in SMTs as the coground form could be considerably improved. In addition, SMTs prepared with FPM showed substantially enhanced swelling properties and appropriate mechanical strength, which are required for SMTs to reside in the stomach for a sufficient period of time and to withstand mechanical stresses caused by gastric motility. The findings of this study would be useful to develop highly swelling GRDDS, capable of releasing hydrophobic drugs with low aqueous solubilities at desired rates.

## Figures and Tables

**Figure 1 pharmaceutics-11-00271-f001:**
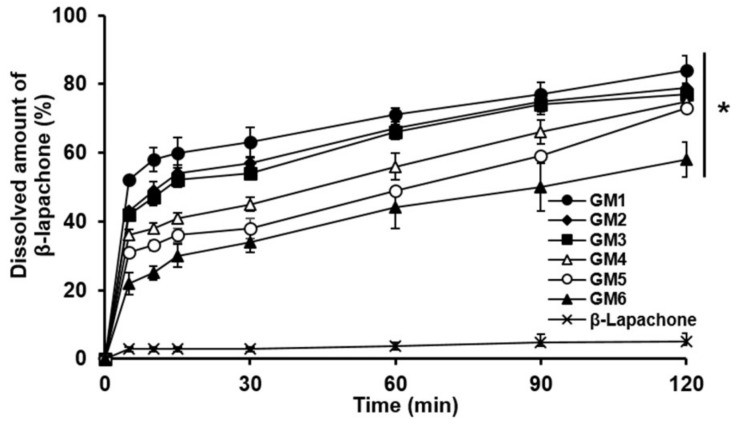
In vitro drug-dissolution profiles of coground mixtures with different compositions evaluated in comparison with raw β-lapachone powder. Values are mean ± SD (*n* = 3). Asterisk denotes a statistical difference at *p* < 0.001 between drug-dissolved amounts evaluated with different coground mixtures and that assessed with raw β-lapachone powder at 120 min.

**Figure 2 pharmaceutics-11-00271-f002:**
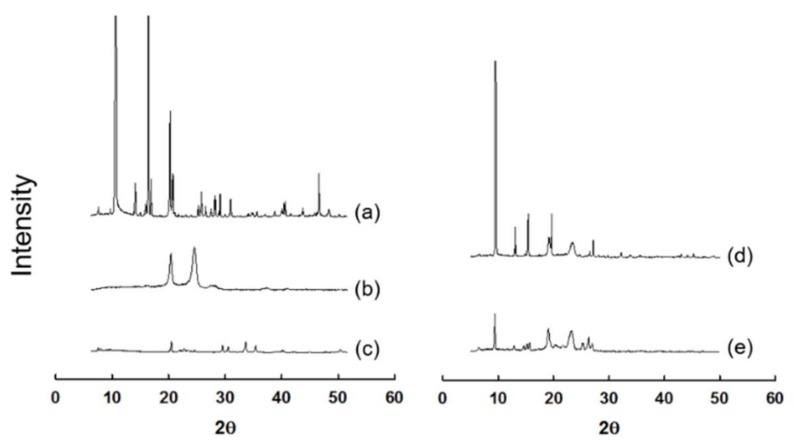
X-ray diffraction (XRD) spectra of (**a**) raw β-lapachone powder, (**b**) poloxamer 407, (**c**) SLS, (**d**) physical mixture of β-lapachone, poloxamer 407, and SLS, and (**e**) coground mixture of β-lapachone, poloxamer 407, and SLS (GM1).

**Figure 3 pharmaceutics-11-00271-f003:**
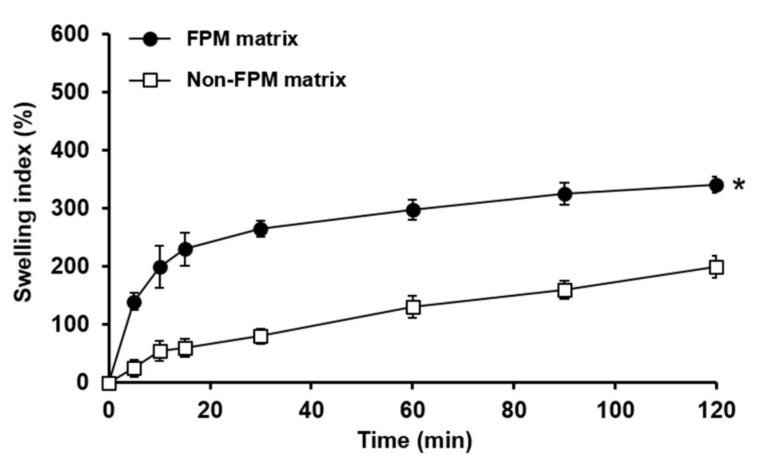
Swelling indices of FPM-based and non-FPM-based matrices evaluated in pH 1.2 simulated gastric fluid for 2 h. The values are mean ± SD (*n* = 3). Asterisk indicates a statistical difference at *p* < 0.001 between the swelling indices of FPM-based and non-FPM-based matrices evaluated at the end of the experiment.

**Figure 4 pharmaceutics-11-00271-f004:**
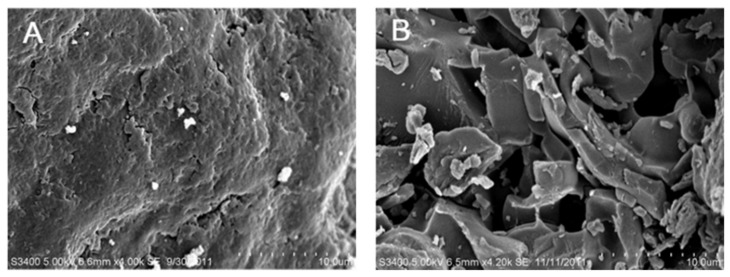
Surface morphology of (**A**) non-FPM-based matrix and (**B**) FPM-based matrix, observed using scanning electron microscopy (SEM).

**Figure 5 pharmaceutics-11-00271-f005:**
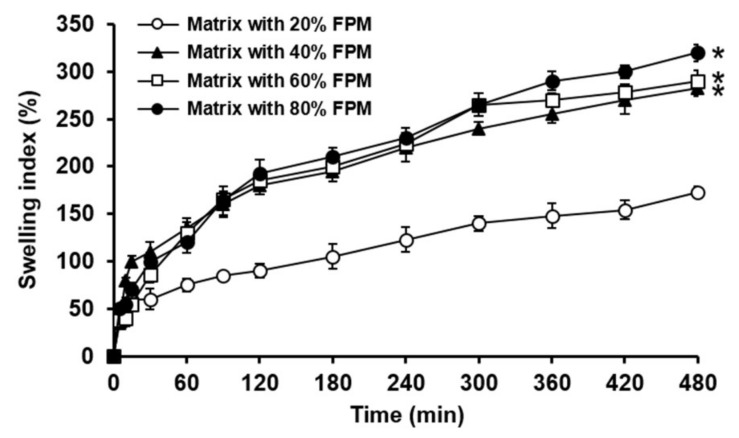
Swelling indices of FPM-based matrices incorporating FPM at different weight fractions of 20%, 40%, 60%, and 80%, assessed for 8 h in pH 1.2 simulated gastric fluid. Values are mean ± SD (*n* = 3). Asterisk means a statistical difference at *p* < 0.001 between the swelling index of FPM-based matrix containing FPM at 20% weight fraction and those of FPM-based matrices incorporating FPM at weight fractions of 40%, 60%, and 80%, evaluated at 8 h.

**Figure 6 pharmaceutics-11-00271-f006:**
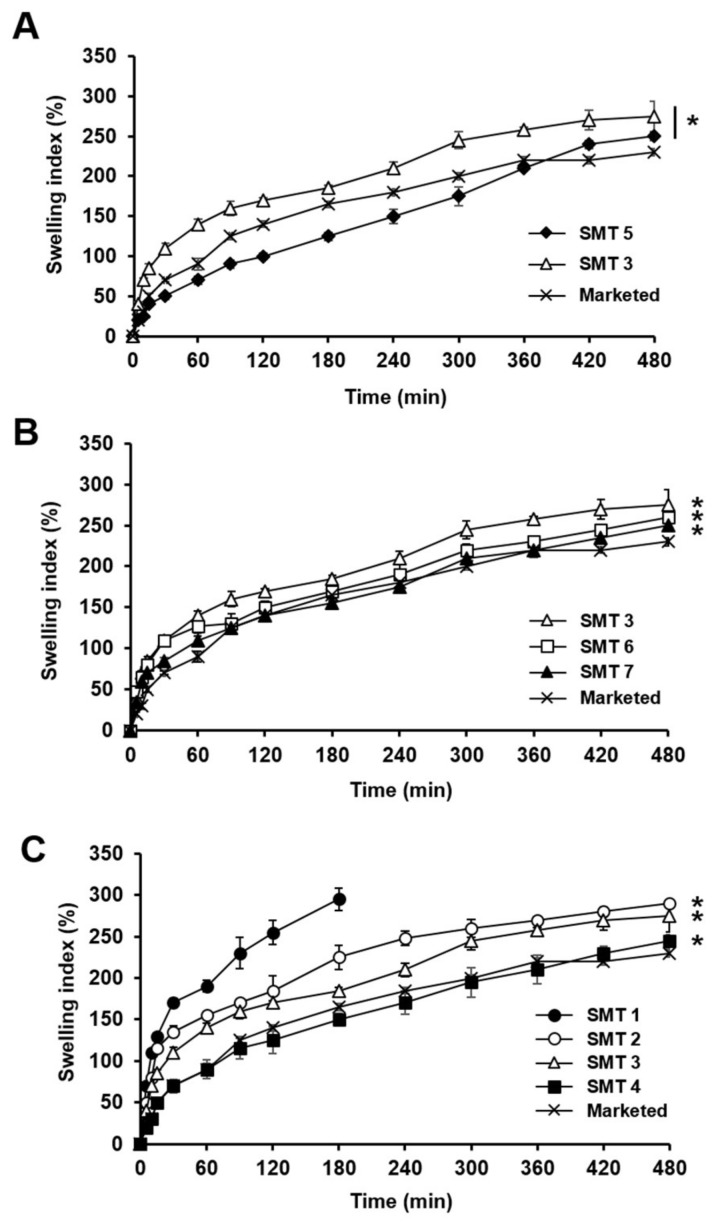
Swelling behaviors of (**A**) SMTs containing FPM (SMT 3) or non-FPM (SMT 5), (**B**) SMTs incorporating PEO of 1000 (SMT 3), 4000 (SMT 6), or 8000 kDa (SMT 7), and (**C**) SMTs with 1000 kDa PEO at different weight fractions of 10% (SMT 1), 15% (SMT 2), 20% (SMT 3), or 30% (SMT 4) assessed in pH 1.2 simulated gastric fluid in comparison with a marketed swelling gastroretentive tablet. Values are mean ± SD (*n* = 3). In Panel A, asterisk indicates a statistical difference at *p* < 0.05 between swelling indices of SMTs 3 and 5, evaluated at 8 h. In Panels B and C, asterisk denotes statistical difference at *p* < 0.05 between the swelling index of the marketed product and those of SMTs assessed at 8 h. As for SMT 1, the swelling index could not be evaluated after 3 h because it was collapsed at 3 h due to its weak mechanical strength.

**Figure 7 pharmaceutics-11-00271-f007:**
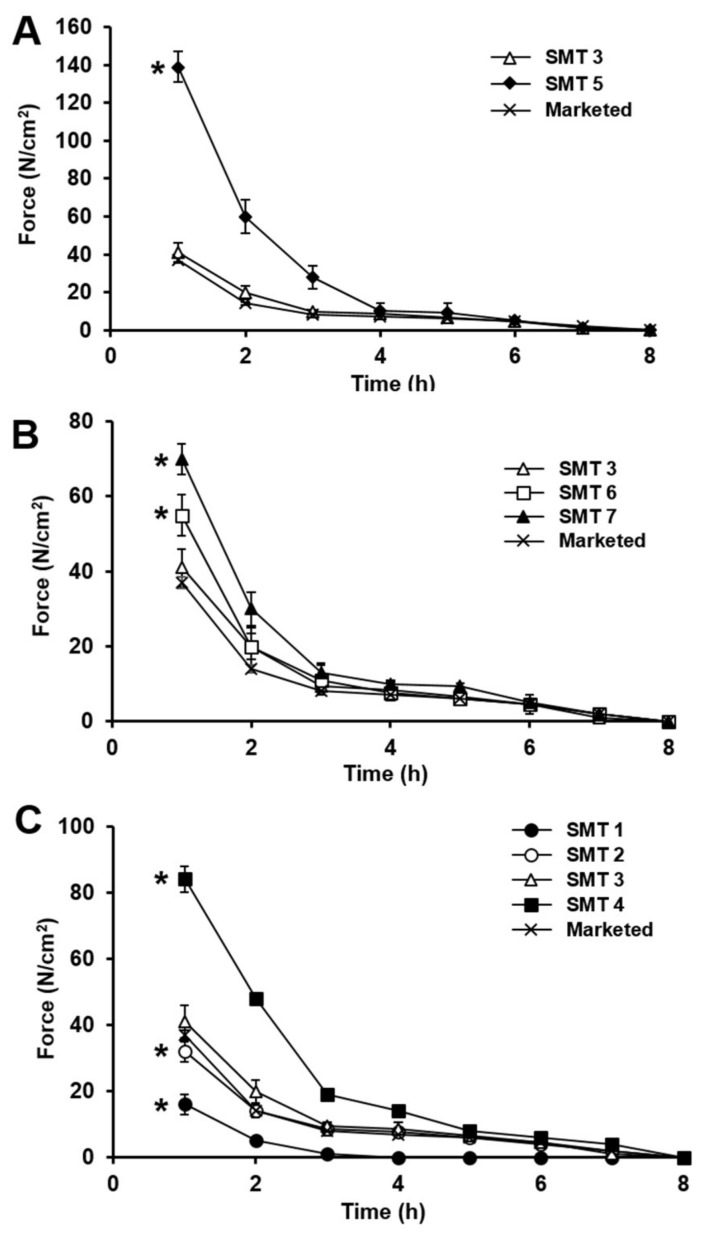
Profiles of mechanical strength of (**A**) SMTs prepared with FPM (SMT 3) or non-FPM (SMT 5), (**B**) SMTs containing PEO of 1000 (SMT 3), 4000 (SMT 6), or 8000 kDa (SMT 7), and (**C**) SMTs with 1000 kDa PEO at different weight fractions of 10% (SMT 1), 15% (SMT 2), 20% (SMT 3), or 30% (SMT 4) evaluated in pH 1.2 simulated gastric fluid for 8 h in comparison with the marketed product. Values are mean ± SD (*n* = 3). Asterisk indicates a statistical difference at *p* < 0.01 between mechanical strengths of the marketed product and SMTs evaluated at 1 h.

**Figure 8 pharmaceutics-11-00271-f008:**
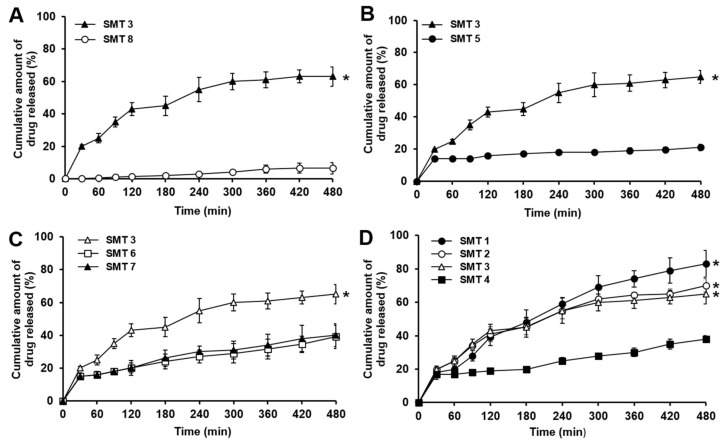
In vitro dissolution profiles of β-lapachone incorporated in (**A**) SMTs prepared with GM1 (SMT 3) or raw β-lapachone powder (SMT 8), (**B**) SMTs containing FPM (SMT 3) or non-FPM (SMT 5), (**C**) SMTs incorporating PEO of 1000 (SMT 3), 4000 (SMT 6), or 8000 kDa (SMT 7), and (**D**) SMTs with 1000 kDa PEO at different weight fractions of 10% (SMT 1), 15% (SMT 2), 20% (SMT 3), or 30% (SMT 4). The dissolution profiles of β-lapachone were assessed for 8 h in pH 1.2 simulated gastric fluid. Values are mean ± SD (*n* = 3). Asterisk indicates a statistical difference at *p* < 0.01 between the dissolved amounts of β-lapachone evaluated at the end of the experiment (8 h) in comparison to the lowest dissolved amount of the drug at the same time point.

**Table 1 pharmaceutics-11-00271-t001:** Compositions of coground mixtures (GM) of β-lapachone, hydrophilic materials, and sodium lauryl sulfate (SLS). Poloxamer 407, poloxamer 188, sucrose, polydextrose, polyethylene glycol 4000, and hydroxypropyl methylcellulose were used as hydrophilic materials. The weight ratio of β-lapachone, each hydrophilic material, and SLS was fixed at 1:3:0.5. All quantities are given in mg.

Ingredients	GM 1	GM 2	GM 3	GM 4	GM 5	GM 6
β-lapachone	40	40	40	40	40	40
Poloxamer 407	120	-	-	-	-	-
Sucrose	-	120	-	-	-	-
Poloxamer 188	-	-	120	-	-	-
Polydextrose	-	-	-	120	-	-
PEG 1 4000	-	-	-	-	120	-
HPMC 2	-	-	-	-	-	120
SLS	20	20	20	20	20	20
**Total**	180	180	180	180	180	180

^1^ Polyethylene glycol; ^2^ Hydroxypropyl methylcellulose.

**Table 2 pharmaceutics-11-00271-t002:** Compositions of swellable matrix tablets (SMTs) of β-lapachone. For the preparation of SMT 1–7, the coground β-lapachone (GM1) was used because it was demonstrated to be the best for improving the dissolution behavior of β-lapachone between the six different coground mixtures shown in Table 1. In the case of SMT 8, raw β-lapachone powder was used.

Ingredients	SMT 1	SMT 2	SMT 3	SMT 4	SMT 5	SMT 6	SMT 7	SMT 8
β-lapachone	5 mg	5 mg	5 mg	5 mg	5 mg	5 mg	5 mg	5 mg
FPM	400 mg	400 mg	400 mg	400 mg	-	400 mg	400 mg	400 mg
Non FPM 1	-	-	-	-	400 mg	-	-	-
PEO 2 (Mw 1000 kDa)	100 mg	150 mg	200 mg	300 mg	200 mg	-	-	200 mg
PEO (Mw 4000 kDa)	-	-	-	-	-	200 mg	-	-
PEO (Mw 8000 kDa)	-	-	-	-	-	-	200 mg	-
SLS	102.5 mg	102.5 mg	102.5 mg	102.5 mg	102.5 mg	102.5 mg	102.5 mg	102.5 mg
Poloxamer 407	115 mg	115 mg	115 mg	115 mg	115 mg	115 mg	115 mg	115 mg
Spray dried lactose	267.5 mg	217.5 mg	167.5 mg	67.5 mg	167.5 mg	167.5 mg	167.5 mg	167.5 mg
Magnesium stearate	10 mg	10 mg	10 mg	10 mg	10 mg	10 mg	10 mg	10 mg
**Total**	1000 mg	1000 mg	1000 mg	1000 mg	1000 mg	1000 mg	1000 mg	1000 mg

^1^ Non-Freeze–Thawed Polymer Mixture (FPM) was prepared using the same ingredients of FPM without the freeze–thaw procedure; ^2^ poly(ethylene oxide).

**Table 3 pharmaceutics-11-00271-t003:** Correlation coefficients (*R*^2^) values for the drug release profiles fitted with various release models.

Formulations	Correlation Coefficients (R^2^) of Drug-Release Kinetics			
	Zero-Order	First-Order	Higuchi	Korsmeyer–Peppas
SMT 1	0.9818	0.9856	0.9926	0.9921
SMT 2	0.9535	0.9881	0.9938	0.9935
SMT 3	0.9088	0.9640	0.9741	0.9798
SMT 4	0.9887	0.9780	0.9376	0.9448
SMT 5	0.9826	0.9825	0.9230	0.8974
SMT 6	0.9964	0.9938	0.9651	0.9724
SMT 7	0.9965	0.9937	0.9767	0.9839
SMT 8	0.9888	0.9833	0.9466	0.9939

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
