# Peer review of "Mechanically Robust Gastroretentive Drug-Delivery Systems Capable of Controlling Dissolution Behaviors of Coground β-Lapachone"

_pharmaceutics, 2019, doi:10.3390/pharmaceutics11060271_

Round 1
Reviewer 1 Report
Dear Editor,
this letter provides my feedback as a reviewer of the paper entitled “Mechanically Robust Matrices Capable of Controlling Dissolution Behavior of Co-ground β-Lapachone in the Stomach” by Jaehwi Lee and coworkers. With this paper the authors evaluate the co-grinding technique and freeze-thaw method in order to prepare amorphous β-lapachone mechanically robust matrix tablets.
In general, the manuscript is well written and easy to follow, while new and novel results are presented for the first time. However, in order for the manuscript to be appropriate for publication in Pharmaceutics, the authors need to adequately address the following issue:
- General comment: The authors should mention in the title and in the abstract that they are evaluating the preparation of a gastro-retentive drug delivery systems (GRDDS).
- line 24: change “for” with “in”.
- poloxamer 188, sucrose, polydextrose, polyethylene glycol 4000, and hydroxypropyl methylcellulose should be mentioned in the materials section.
- line 85: please, clarify which hydrophilic materials.
- line 88: how did the authors achieve powder mixing homogenization, since the vortex mixer is not appropriate to mix powders?
- line 115: how did the author concluded in the 1:1:1 weight ratio of the crospovidone, poly(acrylic acid), and xanthan gum?
- line 135: please delete the phrase “The swelling property of the FPM matrices was assessed”
-line 138: please delete the word “droplets”
- Table 2: please include the weight units in the table (i.e. mg)
- Table 1 shows only six (6) co-ground formulations, however in Table 2 seven (7) SMTs are mentioned (excluding SMT8 which uses raw β-lapachone). Please, clarify.
- lines 174-177: details on the pure API dissolution results are needed. How did the authors performed the dissolution test? Was the API compressed into tablets? Did the use SLS in the prepared tablet? If the authors did not made a pure API tablet, the API was insearted in the dissolution media as a powder, which was the PSD of the API? Was the dissolution test performed under sink conditions?
- lines 196-197: please, explain better the amorphisation claim, since no amorphous halo is observed in the XRD pattern of co-ground mixture.
- lines 208-222: Please delete this paragraph. All info are already noted in the introduction section.
Author Response
RESPONSES TO REVIEWERS’ COMMENTS
First of all, we very much appreciate the reviewers who carefully checked our paper. The reviewers’ comments were carefully studied, responded in a point-by-point manner and reflected in the revised manuscript as described in detail below. In the revised manuscript, the changes were marked in blue highlight.
Reviewer 1
Comments to the author
This letter provides my feedback as a reviewer of the paper entitled “Mechanically Robust Matrices Capable of Controlling Dissolution Behavior of Co-ground β-Lapachone in the Stomach” by Jaehwi Lee and coworkers. With this paper the authors evaluate the co-grinding technique and freeze-thaw method in order to prepare amorphous β-lapachone mechanically robust matrix tablets.
In general, the manuscript is well written and easy to follow, while new and novel results are presented for the first time. However, in order for the manuscript to be appropriate for publication in Pharmaceutics, the authors need to adequately address the following issue:
Specific comments
1) The authors should mention in the title and in the abstract that they are evaluating the preparation of a gastro-retentive drug delivery systems (GRDDS).
[Response] As the reviewer commented, in the revised title and abstract we newly stated that our highly swellable and mechanically robust matrix tablet was developed as a gastro-retentive drug delivery system, which is also shown in below.
[Revised title]
Mechanically Robust Gastro-retentive Drug Delivery Systems Capable of Controlling Dissolution Behavior of Co-ground β-Lapachone
[Revised abstract]
In this study, we aimed to design a highly swellable and mechanically robust matrix tablet (SMT) as gastro-retentive drug delivery systems (GRDDS) capable of improving the dissolution behavior of β-lapachone with a low aqueous solubility. For the preparation of SMTs, the co-grinding technique and freeze-thaw method were used to disperse β-lapachone in SMTs in amorphous state and to enhance the swelling and mechanical properties of SMTs, respectively. As a result, the crystallinity of co-ground β-lapachone incorporated in the SMTs was found to be considerably decreased, and thereby the dissolution rates of the drug in a simulated gastric fluid could be substantially increased. The SMTs of β-lapachone also demonstrated significantly enhanced swelling and mechanical properties compared to those of a marketed product. The reason for this might be because the physically crosslinked polymeric networks with a porous structure were formed in SMTs through the freeze-thaw method. In addition, β-lapachone was gradually released from the SMTs in 6 h. Therefore, SMTs of β-lapachone developed in this study could be usefully used as GRDDS with appropriate swelling and mechanical properties for improving the dissolution behavior of hydrophobic drugs such as β-lapachone.
2) Line 24: change “for” with “in”.
[Response] Done
3) Poloxamer 188, sucrose, polydextrose, polyethylene glycol 4000, and hydroxypropyl methylcellulose should be mentioned in the materials section.
[Response] Done
4) Line 85: please, clarify which hydrophilic materials.
[Response] Done
5) Line 88: how did the authors achieve powder mixing homogenization, since the vortex mixer is not appropriate to mix powders?
[Response] The vortex mixer was used only for rough initial mixing of β-lapachone, hydrophilic materials, and sodium lauryl sulfate. After vortexing the mixtures, a ball mill was used to thoroughly reduce their particle sizes, and promote intensive physical interactions among the constituents of the mixtures. Through the co-grinding technique using the ball mall, β-lapachone dispersed in the co-ground mixtures with a considerably decreased crystallinity was obtained.
6) Line 115: how did the author conclude in the 1:1:1 weight ratio of the crospovidone, poly(acrylic acid), and xanthan gum?
[Response] We would like to thank you very much for the good point. For the preparation of the freeze-thawed polymer mixture (FPM), we selected crospovidone, poly(acrylic acid), and xanthan gum because the hydrophilic polymers have been known to exhibit good swellable properties. Actually, we evaluated the swelling indexes of the FPMs composed of the three different polymers at various weight ratios and as a result, the FPM of the 1:1:1 weight ratio was selected because of its best swelling ability.
In the manuscript, we did not present the data because the data were not considered scientifically significant. We newly explained the reason for choosing the 1:1:1 weight ratio of the hydrophilic polymers as shown on page 3 of the revised manuscript, which is also presented in below.
“The three different hydrophilic polymers were selected because they have been known to exhibit good swellable properties [24-26]. The 1:1:1 weight ratio of the polymers was selected because in our preliminary experiment it demonstrated the best swelling ability among the FPM composed of the polymers at various weight ratios.”
7) Line 135: please delete the phrase “The swelling property of the FPM matrices was assessed”
[Response] Done
8) Line 138: please delete the word “droplets”
[Response] Done
9) Table 2: please include the weight units in the table (i.e. mg)
[Response] Done.
10) Table 1 shows only six (6) co-ground formulations, however in Table 2 seven (7) SMTs are mentioned (excluding SMT8 which uses raw β-lapachone). Please, clarify.
[Response] Among the six different co-ground formulations shown in Table 1, GM1 was chosen because it was demonstrated to be the best for improving the dissolution behavior of β-lapachone as presented in Figure 1. Thereafter, SMTs with seven different compositions (SMT 1-7) were prepared using GM1. To clarify this, we revised the title of Table 2 as shown in below.
Before | After |
Table 2. Compositions of SMTs of β-lapachone. All quantities of the constituents of SMTs are given in mg. | Table 2. Compositions of SMTs of β-lapachone. For the preparation of SMT 1-7, the co-ground β-lapachone (GM1) was used because it was demonstrated to be the best for improving the dissolution behavior of β-lapachone among six different co-ground mixtures shown in Table 1. In case of SMT 8, raw β-lapachone powder was used. |
11) Lines 174-177: details on the pure API dissolution results are needed. How did the authors perform the dissolution test? Was the API compressed into tablets? Did they use SLS in the prepared tablet? If the authors did not made a pure API tablet, the API was inserted in the dissolution media as a powder, which was the PSD of the API? Was the dissolution test performed under sink conditions?
[Response] We very much appreciate the reviewer for the good point. The pure β-lapachone powder was passed through an 80-mesh sieve and used without being tabletized in the dissolution test. We set the sink condition by adding sodium lauryl sulfate to the dissolution media at a concentration of 0.5% (w/v). To add the detailed experimental conditions, we revised the method section as shown on page 3 of the revised manuscript, which is also presented in below.
“In vitro dissolution study of the co-ground mixtures and raw β-lapachone powder was conducted using USP dissolution apparatus II. As for the raw β-lapachone powder, it was passed through an 80-mesh sieve before conducting the dissolution study and used without being tabletized. Samples equivalent to 40 mg of β-lapachone were added to 900 mL simulated gastric fluid (pH 1.2) containing SLS at a concentration of 0.5% (w/v) and stirred at 50 rpm and 37°C.”
12) Lines 196-197: please, explain better the amorphisation claim, since no amorphous halo is observed in the XRD pattern of co-ground mixture.
[Response] We are sincerely grateful to the reviewer for the good point. As the reviewer commented, an amorphous halo was not clearly observed in the XRD pattern of the co-ground mixture (GM1), but GM1 exhibited some small, broad peaks that were not observed from the physical mixture with the same composition to GM1. This result implies that some of β-lapachone could be amorphized by the co-grinding technique. This means also that our technique did not fully cause a perfect amorphization of the drug.
We consider our co-grinding technique was not as efficient as solid dispersion techniques generally employed to obtain amorphous form of a drug within polymeric carriers. However, we could confirm (at least partial) amorphization of β-lapachone only with co-grinding of drug mixture. We employed the co-grinding technique in this study because our co-grinding technique can considerably decrease the crystallinity of β-lapachone only through physical milling and appropriate selection of excipients without using any organic solvents frequently used for preparing solid dispersion. In addition, the co-grinding technique only needs comparatively simple, low-cost equipment and excipients such as a ball mill, surfactants, and polymers, whereas solid dispersion systems require specialized equipment such as hot-melt extruder and spray dryer. Especially spray drying procedure for preparing solid dispersion uses various organic solvents to initially solubilize poorly soluble drugs and polymeric carriers.
Thus, we are certain that the co-grinding technique can be used as an alternative to other pharmaceutical technologies to produce solid dispersion systems for improving the dissolution of hydrophobic drugs.
To reflect this background, we newly added the explanation on this as shown on page 7 of the revised manuscript, which is also presented in below.
“However, an amorphous halo was not clearly observed in the XRD pattern of GM1. This result might be because our co-grinding technique did not fully cause a perfect amorphization of the drug. The co-grinding technique might not be as efficient as solid dispersion techniques generally employed to obtain amorphous form of a drug within polymeric carriers. Nevertheless, the reason for using the co-grinding technique in this study was because the co-grinding technique developed in this study can considerably decrease the crystallinity of β-lapachone only through physical milling and appropriate selection of excipients without using any organic solvents frequently used for solid dispersion. In addition, the co-grinding technique only needs comparatively simple, low-cost equipment and excipients such as a ball mill, surfactants, and polymers, whereas solid dispersion systems require specialized equipment such as hot-melt extruder and spray dryer. Thus, the co-grinding technique could be used as an alternative to other pharmaceutical technologies to produce solid dispersion systems for improving the dissolution of hydrophobic drugs.”
13) Lines 208-222: Please delete this paragraph. All info are already noted in the introduction section.
[Response] Done

Reviewer 2 Report
In this study, the authors designed a highly swellable and mechanically robust matrix tablet (SMT) capable of improving the dissolution behavior of β-lapachone. The co-grinding technique and freeze-thaw method were used to enhance the swelling and mechanical properties. Although this study is interesting, there are some points still need to be addressed and discussed before acceptance.
1. There are many methods to prepare amorphous solid dispersion, why co-grinding was used in this study. It seems that the crystalline drug cannot be completely transformed into amorphous state. So maybe other methods are better to prepare amorphous formulation. Please provide reasonable explanation.
2. The weight ratio of β-lapachone, Poloxamer 407, and SLS in table 1 was quite different from the final tablet composition. The weight ratio of co-grinding mixture was highly recommended to be consistent with SMT.
3. It seems that the mechanical strength and the dissolution rate of the optimized SMT2 were quite similar to those of the commercial tablet. So what is the real advantage of SMT? And the drug in commercial tablet is β-lapachone?
Author Response
RESPONSES TO REVIEWERS’ COMMENTS
First of all, we very much appreciate the reviewers who carefully checked our paper. The reviewers’ comments were carefully studied, responded in a point-by-point manner and reflected in the revised manuscript as described in detail below. In the revised manuscript, the changes were marked in blue highlight.
Reviewer 2
Comments to the author
In this study, the authors designed a highly swellable and mechanically robust matrix tablet (SMT) capable of improving the dissolution behavior of β-lapachone. The co-grinding technique and freeze-thaw method were used to enhance the swelling and mechanical properties. Although this study is interesting, there are some points still need to be addressed and discussed before acceptance.
Specific comments
1) There are many methods to prepare amorphous solid dispersion, why co-grinding was used in this study. It seems that the crystalline drug cannot be completely transformed into amorphous state. So maybe other methods are better to prepare amorphous formulation. Please provide reasonable explanation.
[Response] We very much appreciate the reviewer for the good point. As the reviewer commented, β-lapachone incorporated in the co-ground mixtures was considered to be not fully amorphized based on the result of the XRD analysis.
We consider our co-grinding technique was not as efficient as solid dispersion techniques generally employed to obtain amorphous form of a drug within polymeric carriers. However, we could confirm (at least partial) amorphization of β-lapachone only with co-grinding of the drug mixture. The reason for using the co-grinding technique in this study was because our co-grinding technique can considerably decrease the crystallinity of β-lapachone only through physical milling and appropriate selection of excipients without using any organic solvents frequently used for preparing solid dispersion. The co-grinding technique also only needs comparatively simple, low-cost equipment and excipients such as a ball mill, surfactants, and polymers, whereas solid dispersion systems require specialized equipment such as hot-melt extruder and spray dryer. Especially, the spray drying procedure for preparing solid dispersion uses various organic solvents to initially solubilize poorly soluble drugs and polymeric carriers.
Thus, we are certain that our co-grinding technique can be used as an alternative to other pharmaceutical technologies to produce solid dispersion systems for improving the dissolution of hydrophobic drugs such as β-lapachone.
To reflect this background, we newly added the explanation on this as shown on page 7 of the revised manuscript, which is also presented in below.
“However, an amorphous halo was not clearly observed in the XRD pattern of GM1. This result might be because our co-grinding technique did not fully cause a perfect amorphization of the drug. The co-grinding technique might not be as efficient as solid dispersion techniques generally employed to obtain amorphous form of a drug within polymeric carriers. Nevertheless, the reason for using the co-grinding technique in this study was because the co-grinding technique developed in this study can considerably decrease the crystallinity of β-lapachone only through physical milling and appropriate selection of excipients without using any organic solvents frequently used for solid dispersion. In addition, the co-grinding technique only needs comparatively simple, low-cost equipment and excipients such as a ball mill, surfactants, and polymers, whereas solid dispersion systems require specialized equipment such as hot-melt extruder and spray dryer. Thus, the co-grinding technique could be used as an alternative to other pharmaceutical technologies to produce solid dispersion systems for improving the dissolution of hydrophobic drugs.”
2) The weight ratio of β-lapachone, Poloxamer 407, and SLS in table 1 was quite different from the final tablet composition. The weight ratio of co-grinding mixture was highly recommended to be consistent with SMT.
[Response] We are sincerely grateful to the reviewer for the keen point. Although the co-ground mixture (GM1) exhibited a considerably improved dissolution behavior of β-lapachone compared to pure β-lapachone powder as presented in Figure 1, the drug dissolution evaluated with SMTs containing GM1 was not efficiently enhanced in our preliminary experiment. We considered that this might be because the weight fraction of SLS in SMTs (2%) was considerably lower than that in GM1 (~11%) and thereby the drug recrystallized during mixing the constituents of SMTs and tableting procedure.
Thus, we reoptimized the weight ratios of β-lapachone, poloxamer 407, and SLS in SMTs to inhibit the recrystallization of β-lapachone during the preparation of SMTs. We newly added the explanation on this as shown on pages 12-13 of the revised manuscript, which is also presented in below.
“The weight ratios of SLS and poloxamer 407 used for decreasing the crystallinity of β-lapachone in SMTs were changed from those of GM1 as shown in Table 1 and Table 2. When the same weight ratios of β-lapachone, SLS, and poloxamer 407 to GM1 were used for the preparation of SMTs, the drug dissolution behaviors were not efficiently enhanced (data not shown). The reason for this might be because the weight fraction of SLS in SMTs (2%) was considerably lower than that in GM1 (~11%) and thereby the drug recrystallized during mixing the constituents of SMTs and tableting procedure. Thus, we reoptimized the weight ratios of β-lapachone, SLS, and poloxamer 407 in SMTs to inhibit the recrystallization of β-lapachone during the preparation of SMTs.”
3) It seems that the mechanical strength and the dissolution rate of the optimized SMT2 were quite similar to those of the commercial tablet. So what is the real advantage of SMT? And the drug in commercial tablet is β-lapachone?
[Response] We would like to thank the reviewer very much for the good point. The existing commercially available gastro-retentive tablets are known to exhibit appropriate swelling property and mechanical strength required for prolonged gastric residence and to have an ability to release hydrophilic drugs in a controlled manner. However, the commercial gastro-retentive tablets are not specially formulated to control the dissolution of hydrophobic drugs and therefore, they are not considered to be applicable for β-lapachone with a low aqueous solubility. Thus, it was necessary to develop a swellable gastro-retentive drug delivery system capable of controlling dissolution rates of the poorly soluble drug and maintaining appropriate mechanical strength to resist the gastric contraction.
In this study, we aimed to design a highly swellable, but mechanically robust polymeric matrices that can control the dissolution of β-lapachone co-ground with SLS and another polymer different from those used in the preparation of the matrices. We employed the co-grinding technique to prepare SMTs that mimic solid dispersion systems and can control the dissolution rate of β-lapachone for 8 hours in a swollen state. In addition, the SMTs could be highly swellable and exhibit robust mechanical property owing to their porous polymeric matrices that are physically crosslinked by the freeze-thawing process. We think these aspects are clearly different from commercially available swellable systems.
In this study we used a commercial gastro-retentive tablet that did not contain β-lapachone because gastro-retentive tablets of β-lapachone have not been commercialized yet. For this reason, we unavoidably could not compare the dissolution behaviors of β-lapachone between SMTs and the commercial tablet.

Reviewer 3 Report
The Kim et al manuscript is a valuable research and it is of potential interest for publication in pharmaceutics. In my opinion The following modifications must be done before the publication in order to complete the work and increase the impact and interest of readers toward this research.
-The authors should fit the data of the drug release kinetic with some mathematical models. The correlation coefficients of the models should be compared in order to assess the main phenomena affecting the release kinetic.
-The release is mainly ascribed to the amorphous structure of the drug in the matrix. The authors inquired the crystallinity of the structure by XRD analysis which provides good insight of the crystal structure. However also a DSC analysis should be included. This would not only corroborate the results of the XRD experiment but also provide an hint of the crystallinity at the temperature the release test was performed allowing to better understand the phenomena occurring.
-The introduction should be restyled since it is difficult to read and follow. This is in my opinion very important to communicate the relevance of the present research. My suggestion is to divide introduction in four paragraphs like this:
1) Establish significance and relevance of the topic.
2) Provide the state of the art.
3) Introduce the gap.
4) Present the original work and its novelty.
The present version of the paper mixes this topics all together and is not so smooth to be read.
Also in the introduction the authors should compare the proposed formulation with some of the more advanced ones such us micro and nanoparticles for controlled release a recent easily open access review on this was recently published 10.3390/pharmaceutics10040176.
-The bibliography is quite poor and updated. I suggest the authors to increase the overall number of cited sources well balancing some fundamental and old ones with the more modern and recent ones. Actually the scarce presence of recent publication in the bibliography gives either the idea of poor state of the the art or of a scarce interest of the research community in the present topic.
- In line 49-55 the authors seems to state that incorporation of drug requires the same hydrophilicity of drug and polymer i.e hydrophilic drug in hydrophilic polymer and hydrophobic drug in hydrophobic polymer; and that this is one of the main limitation provided of the current drug delivery system that the present work may overcome. The cited source dates back to 2011. In the last year some (particulate) drug incorporation systems were proposed to incorporate either hydrophobic drug in hydrophilic polymers (10.1016/j.ijpharm.2019.04.053) or hydrophilic drug in hydrophobic polymer 10.3390/polym10101092. In stating the novelty of their work the authors should take also into account these recent findings and mark the difference between the already proposed oral particulate system and the here presented GRDDS.
I strongly suggest the author to put much more attention in revising the introduction section. The way it is written and the choice of reported sources make the reader question the novelty, significance and originality of the work. I believe it is crucial to well communicate the novelty of your work since the research reported is well conducted. More recent development should be cited and critically compared to your work to communicate its significance.
Author Response
RESPONSES TO REVIEWERS’ COMMENTS
First of all, we very much appreciate the reviewers who carefully checked our paper. The reviewers’ comments were carefully studied, responded in a point-by-point manner and reflected in the revised manuscript as described in detail below. In the revised manuscript, the changes were marked in blue highlight.
Reviewer 3
Comments to the author
The Kim et al manuscript is a valuable research and it is of potential interest for publication in pharmaceutics. In my opinion the following modifications must be done before the publication in order to complete the work and increase the impact and interest of readers toward this research.
Specific comments
1) The authors should fit the data of the drug release kinetic with some mathematical models. The correlation coefficients of the models should be compared in order to assess the main phenomena affecting the release kinetic.
[Response] We would like to thank you very much for the good point. To reflect the reviewer’s comment, we fitted our drug release data with mathematical models such as zero-order, first-order, Higuchi, and Korsmeyer-Peppas models and discussed the result as shown on pages 5, 6, 14, and 15 of the revised manuscript, which is also presented in below.
[Page 5, 6 - Method section]
“To investigate the effects of the composition of SMTs on the kinetics of drug release, the drug dissolution data were fitted to four different mathematical equations of drug release: zero-order, first-order, Higuchi, and Korsmeyer-Peppas models.
(Equation 1)
(Equation 2)
(Equation 3)
(Equation 4)
Where Mt is the amount of drug at time t; M0 is the initial amount of drug incorporated in SMTs; Ct is the concentration of drug in dissolution media at time t; C0 is the initial concentration of drug in dissolution media at time; Mt/M∞ is the fraction of drug released at time t; n is the release exponent of Korsmeyer-Peppas model; k0, k1, k2, and k3 are the rate constants of zero-order, first-order, Higuchi model, and Korsmeyer-Peppas, respectively [29].”
[Page 14, 15 – Results and discussion section]
“To examine drug release kinetics from the SMTs, the dissolution data were fitted to zero-order, first-order, Higuchi, and Korsmeyer-Peppas models. Correlation coefficients (R2) for each model evaluated with SMTs of different compositions are presented in Table 3. In general, all the SMTs were found to be well fitted with the zero-order, first-order, Higuchi, and Korsmeyer-Peppas models. However, the Higuchi model can only be applied when the swelling and dissolution of the matrix are negligible [29]. Therefore, we explained the mode of drug release from SMTs by applying zero-order, first-order, and Korsmeyer-Peppas models. The Korsmeyer-Peppas equation is generally used to evaluate the drug release behaviors from polymeric systems when the drug release mechanism is not well elucidated or more than one type of release mechanisms are involved [40]. The release exponent (n) in the Korsmeyer-Peppas equation is used to characterize the drug release mechanism. If the release exponent is equal to or less than 0.45, the drug release is considered to be governed by Fickian diffusion. If the release exponent ranges from 0.45 to 0.89, it indicates anomalous (non-Fickian) transport, which typically involves both diffusion-controlled release and erosion-controlled release. The release exponent of 0.89 denotes Case II (relaxational) transport, and n>0.89 means Super Case II transport [29].
Table 3. Correlation coefficients (R2) values for the drug release profiles fitted with various release models
Formulations | Correlation coefficients (R2) | |||
Zero-order | First-order | Higuchi | Korsmeyer-Peppas | |
SMT 1 | 0.9818 | 0.9856 | 0.9926 | 0.9921 |
SMT 2 | 0.9535 | 0.9881 | 0.9938 | 0.9935 |
SMT 3 | 0.9088 | 0.9640 | 0.9741 | 0.9798 |
SMT 4 | 0.9887 | 0.9780 | 0.9376 | 0.9448 |
SMT 5 | 0.9826 | 0.9825 | 0.9230 | 0.8974 |
SMT 6 | 0.9964 | 0.9938 | 0.9651 | 0.9724 |
SMT 7 | 0.9965 | 0.9937 | 0.9767 | 0.9839 |
SMT 8 | 0.9888 | 0.9833 | 0.9466 | 0.9939 |
SMT 3 with FPM had the best fit with Korsmeyer-Peppas model, whereas SMT 5 with non-FPM exhibited the best fit with the zero-order kinetic. As for SMT 5, its non-porous structure caused by non-FPM could not efficiently absorb water compared to SMT 3 with a porous structure, thereby delaying the dissolution of drug. The n value of SMT 3 was determined to be between 0.45 and 0.89 (0.4533), indicating that the drug release was governed by both drug diffusion and matrix erosion. The reason for this might be because the presence of FPM in SMT 3 caused a fast swelling rate, thereby promoting the erosion of the polymeric matrix of SMT 3 and the drug release.
The effect of the molecular weight of PEO on the drug release kinetics from SMTs was then examined. As aforementioned, SMT 3 with 1,000 kDa PEO presented a drug release kinetic well fitted to the Korsmeyer-Peppas model, whereas SMT 6 with 4,000 kDa PEO and SMT 7 with 8,000 kDa PEO exhibited the best fit with zero-order model. The reason for SMT 6 and SMT 7 showing zero-order kinetics might be because PEO of a higher molecular weight could be largely entangled each other and form a physically dense structure, thereby delaying the diffusion of water molecules into the matrices and leading to the slow drug release from the SMTs.
The impact of the weight fraction of PEO in SMTs on the drug release behaviors was then explored. SMT 1-3 showed the best fit with the Korsmeyer-Peppas model, whereas SMT 4 presented the best fit with the zero-order kinetic. The n value determined from SMT1-3 ranged from 0.45 to 0.89. The n value also decreased with increasing the weight fraction of PEO in SMTs. This might be because the higher weight fraction of PEO might formed more condensed structures, and consequently the drug release was mainly governed by drug diffusion rather than by matrix erosion. In case of SMT 4, the highest fraction of PEO in the tablet was supposed to inhibit the infiltration of water into the polymeric matrix and delay the drug release. Thus, SMT 4 showed the zero-order drug release kinetic.”
2) The release is mainly ascribed to the amorphous structure of the drug in the matrix. The authors inquired the crystallinity of the structure by XRD analysis which provides good insight of the crystal structure. However, also a DSC analysis should be included. This would not only corroborate the results of the XRD experiment but also provide a hint of the crystallinity at the temperature the release test was performed allowing to better understand the phenomena occurring.
[Response] We very much appreciate the reviewer for the keen point. We fully agree with the reviewer about the importance of the DSC analysis for supporting the result of XRD analysis and understanding the effect of temperature on the crystallinity of drugs.
Before we evaluated the crystallinity of β-lapachone contained in the co-ground mixtures, we studied several literatures that performed the DSC analysis of β-lapachone molecularly dispersed in hydrophilic polymer-based matrices [1, 2]. In the literatures, pure β-lapachone powder in crystalline state showed only single sharp endothermic peak at 157°C caused by the drug melting, and the sharp endothermic peak was demonstrated to disappear when β-lapachone was dispersed in the formulations in amorphous state. However, both pure β-lapachone powder and the drug molecularly dispersed in the formulations did not exhibit any other endothermic or exothermic peaks.
[1] Dos Santos, K.M.; Barbosa, R.M.; Vargas, F.G.A.; de Azevedo, E.P.; Lins, A.C.D.S.; Camara, C.A.; Aragão, C.F.S.; Moura, T.F.L.E.; Raffin, F.N. Development of solid dispersions of β-lapachone in PEG and PVP by solvent evaporation method. Drug Dev Ind Pharm 2018, 44, 750-756.
[2] Cunha-Filho, M.S.; Dacunha-Marinho, B.; Torres-Labandeira, J.J.; Martínez-Pacheco, R.; Landín, M. Characterization of beta-lapachone and methylated beta-cyclodextrin solid-state systems. AAPS PharmSciTech 2007, 8, E60.
Based on this, in this study we conducted only XRD analysis to evaluate the crystallinity of β-lapachone incorporated in the co-ground mixture and did not performed the DSC analysis. We newly added the explanation on this as shown on page 7 of the revised manuscript, which is also presented in below.
“Differential scanning calorimetry (DSC) analysis might be necessary to support the result of the XRD analysis and to understand the effect of temperature on the crystallinity of β-lapachone incorporated in GM1. The reason why the DSC analysis was not performed in this study was because previously published literatures demonstrated that pure β-lapachone powder in crystalline state did not exhibit any endothermic or exothermic peaks other than its characteristic endothermic peak at 157°C caused by melting of the drug [31,32]. In addition, β-lapachone molecularly dispersed in hydrophilic polymer-based matrices was also found to show no endothermic or exothermic peaks over temperature ranges generally used for the DSC analysis. Based on this, in this study only XRD analysis was conducted to evaluate the crystallinity of β-lapachone incorporated in the co-ground mixture.”
3) The introduction should be restyled since it is difficult to read and follow. This is in my opinion very important to communicate the relevance of the present research. My suggestion is to divide introduction in four paragraphs like this:
(1) Establish significance and relevance of the topic.
(2) Provide the state of the art.
(3) Introduce the gap.
(4) Present the original work and its novelty.
The present version of the paper mixes this topics all together and is not so smooth to be read.
[Response] We very much appreciate the reviewer for the good point. To reflect the reviewer’s comment, we revised the introduction section as shown on pages 1-2 of the revised manuscript and marked with blue color. The revised introduction section is now composed of four paragraphs as the reviewer suggested.
4) Also in the introduction the authors should compare the proposed formulation with some of the more advanced ones such us micro and nanoparticles for controlled release a recent easily open access review on this was recently published 10.3390/pharmaceutics10040176.
[Response] To reflect the reviewer’s comment, we stated the limitation of micro-/nano-particulate systems for controlled drug release in achieving efficient oral absorption of drugs with narrow absorption sites, unstable or poorly soluble natures in alkaline pH, and explained the necessity of gastro-retentive drug delivery systems as presented on page 1 of the revised manuscript, which is also shown in below. We also newly cited the literature the reviewer suggested in the revised manuscript.
“To maximize the advantages of oral drug administration, various micro-/nano-particulate systems for controlled drug release have been extensively explored [1]. However, such micro-/nano-particulate systems may not be effective in obtaining high oral bioavailability of drugs with narrow absorption sites, unstable or poorly soluble natures in alkaline pH [2]. To increase the oral bioavailability of such drugs, gastro-retentive drug delivery systems (GRDDS) can be useful. GRDDS can stay in the stomach and release drugs for a prolonged period, thereby enhancing the oral bioavailability and therapeutic efficacy of the drugs [3].”
5) The bibliography is quite poor and updated. I suggest the authors to increase the overall number of cited sources well balancing some fundamental and old ones with the more modern and recent ones. Actually the scarce presence of recent publication in the bibliography gives either the idea of poor state of the art or of a scarce interest of the research community in the present topic.
[Response] We would like to thank you very much for the good point. As the reviewer commented, we newly cited recently published literatures in the revised manuscript. In the reference list, the changes are marked in blue color.
6) In line 49-55, the authors seems to state that incorporation of drug requires the same hydrophilicity of drug and polymer i.e. hydrophilic drug in hydrophilic polymer and hydrophobic drug in hydrophobic polymer; and that this is one of the main limitation provided of the current drug delivery system that the present work may overcome. The cited source dates back to 2011. In the last year some (particulate) drug incorporation systems were proposed to incorporate either hydrophobic drug in hydrophilic polymers (10.1016/j.ijpharm.2019.04.053) or hydrophilic drug in hydrophobic polymer 10.3390/polym10101092. In stating the novelty of their work the authors should take also into account these recent findings and mark the difference between the already proposed oral particulate system and the here presented GRDDS.
[Response] We very much appreciate the reviewer for the keen point. Our purpose of the statements the reviewer pointed out was to mention that existing gastro-retentive drug delivery systems, which are generally composed of hydrophilic polymers, have not been specially formulated to solubilize hydrophobic drugs and enhance the drug dissolution behavior and thus they have limitation in controlling the dissolution behavior of hydrophobic drugs.
To avoid confusion and clearly state this, we revised the sentences the reviewer pointed out and newly cited relevant literatures that were recently published as shown on page 2 of the revised manuscript, which is also presented in below.
“In addition, most existing GRDDS have not been specially formulated to solubilize hydrophobic drugs and improve the dissolution behavior of the drugs and therefore, they have been generally used only for hydrophilic drugs [12-15].”
The reason for not citing the literatures the reviewer mentioned was because we consider that the literatures are related to efficient drug encapsulation in nanoparticles composed of polymers with different hydrophilic/hydrophobic property from drugs and thus they are not closely associated with our purpose.
7) I strongly suggest the author to put much more attention in revising the introduction section. The way it is written and the choice of reported sources make the reader question the novelty, significance and originality of the work. I believe it is crucial to well communicate the novelty of your work since the research reported is well conducted. More recent development should be cited and critically compared to your work to communicate its significance.
[Response] We fully agree with the reviewer about the importance of the way the introduction is written and references for presenting the novelty and significance of our work. With this in mind, we did our best to revise the introduction section. Again, we sincerely appreciate the reviewer for the good point.

Round 2
Reviewer 2 Report
The authors have addressed the points raised by the reviewer and revised the manuscript critically. The reviewer think the revised manuscript is suitable for publication.
Reviewer 3 Report
The Authors seriously and cerefully took into consideration the point I arised the revised version.
I therefore suggest the paper to be suitable for pubblication in pharmaceutics.